# Systemic Disease Associations in a Cohort of Hispanic Patients with Scleritis

**DOI:** 10.3390/jcm12051969

**Published:** 2023-03-02

**Authors:** Cristina Arruza, Guillermo A. Requejo Figueroa, Javier Colón, Estefania Ramirez Marquez, Guillermo Puebla, Daniela Ortega, Mariella Pappaterra Rodriguez, José López Fontanet, Alexandra Colón, Sofía C. Ayala Rodríguez, Erick Rivera Grana, Frances Marrero, Emilio Báez, Carmen Santos, Armando L. Oliver

**Affiliations:** 1Department of Ophthalmology, Medical Sciences Campus, University of Puerto Rico School of Medicine, San Juan, PR 00921, USA; 2Universidad Central del Caribe School of Medicine, Bayamón, PR 00960, USA; 3Ponce Health Sciences University School of Medicine, Ponce, PR 00716, USA

**Keywords:** scleritis, immune-mediated systemic disease, infectious disease, Hispanics

## Abstract

(1) Purpose: A patient with scleritis may have an associated systemic disease, which is often autoimmunological and seldom infectious in origin. The data regarding such associations in Hispanic populations are scarce. Therefore, we evaluated the clinical characteristics and systemic-disease associations of a cohort of Hispanic patients with scleritis. (2) Methods: A retrospective review of the medical records (January 1990–July 2021) of two private uveitis practices in Puerto Rico was performed. Clinical characteristics and systemic-disease associations observed either at presentation or diagnosed as a consequence of the initial workup were recorded. (3) Results: A total of 178 eyes of 141 patients diagnosed with scleritis were identified. An associated autoimmune disease was present in 33.3% of the patients (rheumatoid arthritis, 22.7%; Sjögren’s syndrome, 3.5%; relapsing polychondritis, 2.8%; sarcoidosis, 1.4%; systemic lupus erythematosus, 1.4%; and systemic vasculitis, 0.7%). An associated infectious disease was present in 5.7% of the patients (2.13%, syphilis; 1.41%, herpes simplex; 1.14%, herpes zoster; and 0.71%, Lyme disease). One patient had all-trans retinoic-acid-associated scleritis. Statistical analysis revealed that patients with nodular anterior scleritis were less likely to have an associated immune-mediated disease (OR: 0.21; *p* = 0.011). (4) Conclusion: Rheumatoid arthritis was the most common systemic autoimmune disease association, while syphilis was the most common infectious disease associated with scleritis patients. Our study suggests that patients with nodular scleritis have a lower risk of having an associated immune-mediated disease.

## 1. Introduction

Scleritis is an inflammatory ocular disorder within the scleral wall of the eye [1]. It has been repeatedly reported that a scleritis diagnosis is most often associated with a systemic disease [1,2,3]. Previous studies have reported that 40% to 50% of all patients with scleritis have an associated infectious or autoimmune disease; 5% to 10% of them have an infectious disease as the origin, while the other 30% to 40% have an associated immune-mediated systemic disease [4]. Knowing the type of scleritis and whether an associated systemic disease is present may help healthcare providers identify the optimal treatment measures. For instance, an infectious disease requires antibiotic or antiviral therapy, while an autoimmune-mediated disease requires systemic therapy with non-steroidal anti-inflammatory drugs (NSAIDs), corticosteroids, or immunosuppressive drugs [1,4].

The diagnosis of scleritis is made clinically, based on the patient’s comprehensive medical history, clinical exam, and seldom used ancillary imaging techniques, such as B-scan ultrasound [2]. The most common classification system for scleritis was published in 1976 by Watson and Hayreh from Moorfields Eye Hospital in London, UK [3]. This system is based on the anatomical site and clinical appearance of the existing inflammation and the initial presentation [3]. Scleritis can involve the anterior or the posterior sclera [2,4]. Anterior scleritis is further classified as diffuse, nodular, or necrotizing [2,4]. Posterior scleritis involves the sclera posterior to the insertion of the recti muscles, and B-scan ultrasonography may be used for its diagnosis [1]. Ocular complications commonly seen in patients with scleritis include a decrease in vision, anterior uveitis, peripheral keratitis, and ocular hypertension [1].

To the best of our knowledge, there are no reports in the medical literature characterizing a cohort of scleritis patients of Hispanic origin in Latin America, including Puerto Rico. The aim of our study was to describe the demographic and clinical characteristics (upon presentation) of a cohort of Hispanic scleritis patients living in Puerto Rico.

## 2. Methods

We performed a retrospective review of medical records generated from January 1990 through July 2021; file records were related to scleritis patients from two private uveitis practices and the pertinent outpatient clinics of the Department of Ophthalmology of the University of Puerto Rico School of Medicine. The scleritis diagnoses of these patients were made by certified uveitis specialists who assessed a combination of characteristic clinical findings, history of present illness, and past medical history. Scleritis was often differentiated from episcleritis by the application of 10% phenylephrine drops. Scleritis included edema that affected the episcleral and scleral tissues and congestion of superficial and deep scleral vessels; of the two, only congestion of deep scleral vessels remains after the application of phenylephrine drops. Scleritis was classified as diffuse anterior, nodular anterior, necrotizing, or posterior, according to the definitions set forth by Watson and Hayreh. All the patients underwent a comprehensive physical examination and an extensive systemic workup to identify scleritis-associated conditions such as rheumatoid arthritis, systemic vasculitis, systemic lupus erythematosus (SLE), syphilis, and viral etiologies. The ascertainment of an associated immune-mediated or infectious disease at a given patient’s presentation was made by a review of that individual’s clinical history, an examination, and a clinical diagnosis supported by ancillary laboratory testing. Consultations by an internal medicine specialist or rheumatologist were sought when needed, on a case-by-case basis. The previously identified conditions were included in each patient’s history if that person had been diagnosed before the initial visit (within 1 month) or because of the initial medical workup order for the scleritis diagnosis. Patient demographics such as age, sex, race, and ethnicity were recorded.

Furthermore, the data compilation included visual acuity (VA) and intraocular pressure (IOP) at presentation, ocular complications such as keratitis and uveitis, past medical (including ocular) history, and a laboratory workup. This workup included immunological and serological tests that looked at HLA-B27 positivity, antineutrophil cytoplasmic antibody levels (ANCA), rheumatoid factor (RF), antinuclear antibody (ANA), cyclic citrullinated peptide, the Sjögren’s antibodies SS-A (anti-Ro) and SS-B (anti-La), rapid plasma reagin, fluorescent treponemal antibody absorption (FTA-ABS), purified protein derivative, QuantiFERON-Tb, herpes simplex virus (HSV)-1 and -2 IgG and IgM, and herpes zoster IgG and IgM; a chest X-ray was performed as well.

The data obtained from the review of the medical records of the patients with scleritis who met the study criteria were prospectively entered into a new database for analysis. The database included demographic and clinical data, past medical history, associated immune-mediated systemic and/or infectious diseases, and laboratory workups. Descriptive statistical analysis was performed using the Microsoft Excel software program. The frequencies of clinical and demographic variables were tabulated to facilitate said analysis.

Fisher’s exact test was utilized to analyze significant risk factors for infectious disease or immune-mediated disease using the statistical software program JMP. A *p* value of 0.05 was considered significant. The corresponding confidence intervals and odds ratios of each potential risk factor were included in the analysis. The University of Puerto Rico Medical Sciences Campus Institutional Review Board reviewed and approved this protocol.

## 3. Results

A total of 141 patients with scleritis were identified over a 31-year period. Patient demographic and clinical characteristics have been summarized (see Table 1). The patients ranged in age from 10 to 86 years; the median age at presentation was 53 years. Of those 141 patients, 99 were female (70.2%). All the patients identified themselves as Hispanic. The total number of eyes with scleritis was 178. Patients with only one eye affected accounted for 73.8% of the group, while 37 patients (26.2%) had bilateral scleritis. The most common type of scleritis in the patients at presentation was diffuse anterior (72.3%), which was followed by nodular anterior scleritis (19.1%), posterior scleritis (7.8%), and necrotizing scleritis (4.3%).

Table 2 shows the analysis for the number of eyes diagnosed with scleritis. The types of scleritis by eye followed the same tendency as the type of scleritis in patients. There were four patients who each had two different types of scleritis in their two eyes. For example, one patient had posterior scleritis in the right eye and diffuse anterior and posterior scleritis in the left eye. The other patients had a combination of diffuse anterior with nodular anterior or diffuse anterior with posterior scleritis. Almost 14% of the eyes with scleritis also had uveitis, which is inflammation of the uvea (the middle layer of tissue in the eye wall). Only 1.74% of eyes with scleritis were also affected with keratitis (inflammation of the cornea). Visual acuity was equal to or less than 20/200, including counting fingers, hand motion, and no light perception, in 11.6% of the eyes with scleritis.

Nearly 40% of the patients with a scleritis diagnosis also had an immune-mediated systemic or infectious disease. Table 1 demonstrates the most common conditions seen in this group of patients. Rheumatoid arthritis was the most common immune-mediated systemic disease, followed by Sjögren’s syndrome and relapsing polychondritis (RP). The two patients with seronegative spondyloarthropathies had a diagnosis of psoriatic arthritis. The two patients with SLE and the two patients with sarcoidosis were all females. Only a single 70-year-old female patient had been diagnosed with granulomatosis with polyangiitis, a form of systemic vasculitis (Wegener’s). The other immune-mediated diseases included IgA nephropathy, multiple sclerosis, juvenile idiopathic arthritis (JIA), and psoriasis. One patient, who was being treated for acute promyelocytic leukemia, suffered from a form of scleritis that had been brought on by the medication that she was taking, all-trans retinoic acid (ATRA).

The presence of infectious diseases in patients with scleritis has also been discussed in the literature. In our patient population, nearly 6% of the patients had an active infectious disease at the time of their presentation of scleritis. Two patients had an untreated infection of syphilis and positive FTA-ABS. The third patient with a syphilis diagnosis had a negative FTA-ABS test but a diagnosis of HIV, which impacts the possible treatment for scleritis. Several patients had scleritis associated with either herpes simplex or herpes zoster ophthalmicus; in patients with herpes, the type was confirmed.

The risk factors for infectious and immune-mediated diseases in patients with scleritis are shown in Table 3. Nodular anterior scleritis (OR = 0.21; *p* = 0.011) was the only type of scleritis seen less often in patients with an immune-mediated disease. No other significant risk factors were found.

## 4. Discussion

Our study focused on the characteristics at presentation of 141 Hispanic patients with scleritis. In our cohort, all the patients identified as Hispanic, were diagnosed with a type of scleritis and evaluated for an associated systemic disease of autoimmune or infectious origin. Our results were similar to those observed in several previous series in terms of both demographic characteristics (sex and age) of our sample population and the bilaterality of their condition at presentation.

In our population, females were more likely than males to be affected (70.2%). This is similar to what has been reported by other studies, in which female patients made up 60% to 74% of the total sample. In addition, our population was comparable to that of Akpek et al., in whose study 70.3% of the sample identified as women [2,4,5,6]. The median age at presentation was 53 years, consistent with the range of 47 to 60 years observed in previous reports [2,4,5,6]. Approximately 26% of the patients in our cohort presented with bilateral scleritis, while in the Jabs et al. and Akpek et al. series, it was close to half of the population [2,4]. Nevertheless, in other studies, bilateral scleritis was seen in 25.5% to 32% of the patients [5,7,8]. Scleritis was further divided into four types. In our study, most of the patients presented with diffuse anterior scleritis (72.3%), followed by nodular anterior scleritis (19.1%), posterior scleritis (7.8%), and necrotizing scleritis (4.3%). Likewise, Sainz de la Maza et al., Lin et al., and Akpek et al. reported the same order of prevalence of the different types of scleritis [1,4,8]. On the other hand, Berchicci et al. reported higher rates of both nodular anterior scleritis (33%) and necrotizing anterior scleritis (7%) and lower rates of both diffuse anterior (58%) and posterior scleritis (2%), further emphasizing how important it is to analyze the types of scleritis seen in specific populations [5]. Jabs et al. also had a population with more necrotizing anterior scleritis than posterior scleritis [2].

Ocular complications at presentation, such as uveitis, keratitis, decreased VA, and increased IOP (greater than 21), were measured. It is difficult to compare our population with the populations of other series because each study has its own time frame for evaluating ocular complications. Furthermore, different studies define decreases in VA in their own manner. For example, Sainz de la Maza et al. and Jabs et al. both defined decreased VA as loss of two Snellen lines at the end of the follow-up period. Compared to what was observed at the initial presentation in Sainz de la Maza’s study, 15.8% of the patients had experienced this loss by the end of the follow-up period, while in Jabs et al.’s study, it was 15.9% of the patients [1,2]. Our study focused only on those affected eyes that had a best-corrected VA of 20/200 or worse at presentation (11.8%) to represent how VA was affected. Most of the patients with scleritis (in our sample) did not experience a decrease in vision, denoting that being afflicted with scleritis does not mean that a decrease in VA will occur. Uveitis, keratitis, and increased IOP were also reported as complications in affected eyes, making it difficult to compare between series.

Though we searched the medical literature, we could not identify any other studies that had done a retrospective analysis of patients who identified as Hispanic and had a diagnosis of scleritis. However, Lin et al. performed a study on systemic autoimmune disease in patients with idiopathic scleritis; the study included a population in which 16% of the patients identified as Hispanic [8]. It was noted that being Hispanic is a statistically significant risk factor for presenting with a previously diagnosed autoimmune condition in patients with scleritis [8]. Furthermore, the data regarding patient demographics, disease complications, and systemic disease associations for scleritis in the Puerto Rican population is scarce. Herein, we evaluated a cohort of Hispanic patients living in Puerto Rico, all of whom had a diagnosis of scleritis, to gather further insight into the peculiarities of this disease within our population and its relation to immune-mediated systemic diseases and infectious diseases.

As stated above, previous studies have reported that 40% to 50% of patients with scleritis have an associated infectious or immune-mediated systemic disease; 5% to 10% of said patients have an infectious disease as the origin, and 30% to 40% have an associated immune-mediated systemic disease [4]. Our study follows the same pattern, in that 39% of our patients had scleritis and an associated disease at presentation. Furthermore, approximately 33% of our patients had an immune-mediated systemic disease compared to 37% and 39.2% of the patients who took part in the series of Akpek et al. and Jabs et al., respectively [2,4]. In our series, rheumatoid arthritis (22.7%) was the most common such disease, as was also the case with the respective cohorts of Akpek et al. (15.2%) and Jabs et al. (17.5%) [2,4]. The high prevalence of scleritis patients with rheumatoid arthritis supports the notion that scleritis is a complication of rheumatoid disease; knowledge of this relationship will aid in the monitoring of such patients as part of their long-term care [9]. In the patients in their sample group (compared to a control group), Jayson and Jones found a significant association between rheumatoid arthritis and scleritis [10]. The exacerbation of scleritis often coincided with flare-ups or complications of rheumatoid arthritis, further proving how important it is to monitor patients and treat them with corticosteroids or immunosuppressive therapy as needed to avoid such flare-ups, and Sainz de la Maza et al. found that rheumatoid arthritis patients with scleritis were older than patients with idiopathic scleritis were; this team noted as well that the former patients more frequently suffered from necrotizing scleritis, decreased vision, and peripheral ulcerative keratitis than did the latter [11]. Screening for immune-mediated systemic diseases in patients with a scleritis diagnosis using laboratory markers and clinical histories is crucial to understand how these conditions present in our population, avoiding further complications, and deciding on appropriate treatments. Lin et al. highlight the importance of ANCA and RF as the most common rheumatoid disease markers [8]. Given the prevalence of rheumatoid arthritis in Hispanic patients with scleritis, initial laboratory workup should include these markers.

Other immune-mediated systemic diseases seen in our population include Sjögren’s syndrome, RP, SLE, sarcoidosis, and granulomatosis with polyangiitis. More recent studies in the field have reported an association between Sjögren’s syndrome and scleritis [6,12]. In our study, 3.5% of the patients with scleritis at presentation also had Sjögren’s syndrome; this is similar to what was found by both Berkenstock (4.5%) and Jan et al. (4.10%). Jan et al. found that after controlling for other sociodemographic factors and possible comorbidities, patients with Sjögren’s syndrome had a significantly higher risk of developing scleritis than did patients with other known risk factors (but not Sjögren’s syndrome) [6]. Though recent studies have reported an association between Sjögren’s syndrome and scleritis, population-based epidemiological data derived from Hispanics and concerning the strength of this association remain lacking, which limits proper management and treatment. Jabs et al., Akpek et al., and Thorne et al. series did not report this association, which opens the question about Sjögren’s syndrome and its prevalence, specifically in the Hispanic population. These patients can present with scleritis as the initial ocular complication, accompanied by undiagnosed Sjögren’s syndrome. Furthermore, the association highlights the importance of including anti-Ro, anti-La, RF, and ANA markers in initial systemic workups to decide on disease management to prevent further complications.

We observed RP in 2.8% of our patients, while 1.6% and 3.1% of the patients in Akpek et al.’s and Jabs et al.’s studies were reported to suffer from RP and scleritis, respectively [2,4]. It has been reported that RP is a disease of intermediate severity; therefore, it is important to diagnose it when it is present in patients with scleritis to avoid further ocular complications [13]. This association is important to highlight in our series. Other studies have shown that a decrease in vision is observed more often in scleritis patients with RP than in scleritis patients with some other systemic immune-mediated diseases. [14]. Furthermore, it has been shown that most scleritis patients require cytotoxic immunosuppressive chemotherapy to treat their disease [15]. Other immune-mediated diseases reported in different series include SLE and systemic vasculitis. Although in our series only 1.4% of the scleritis patients had SLE and 0.7% had systemic vasculitis, Jabs et al. reported a higher percentage of patients with scleritis and one of these associated diseases: 4.1% of said patients had SLE, and 7.2% to 9.1% of them had systemic vasculitis [2,4]. Several factors may have contributed to this variation, including the differences in prevalence and early diagnosis in our population. This could be due to them having other manifestations and thus being rapidly treated to control and avoid ocular complications. Another interesting finding in our study, also seen in the Berkenstock series, was a case of JIA with scleritis, which was reported in the “other” category of patients with an immune-mediated systemic disease (Table 1) [12].

While multiple series have reported expected immune-mediated systemic diseases such as rheumatoid arthritis, RP, SLE, and vasculitis, our series also includes patients with scleritis that had a sarcoidosis diagnosis. Dursun et al. discuss how scleritis was the first manifestation of sarcoidosis in two patients, highlighting the importance of identifying this association in our population [16]. In those cases, conjunctival biopsies were necessary to arrive at the diagnosis of sarcoidosis. It is also important to note that, in those particular cases, the patients responded to systemic corticosteroid treatment [16]. The finding of sarcoidosis in a Hispanic population with a diagnosis of scleritis adds value to the initial workup performed and helps healthcare providers determine the best treatment to alleviate the symptoms.

While series such as those of Akpek et al. and Jabs et al. found herpes zoster ophthalmicus (4.5% to 5.2%) and herpes simplex (1.6% to 2.1%) to be the most common infectious diseases in patients with scleritis, our patients most commonly presented with syphilis as the associated infectious disease [2,4]. It is important to highlight the stronger prevalence of this condition in our Hispanic population, as studies have shown that neither topical nor oral corticosteroid therapy improves symptoms, while ocular inflammation tends to resolve after the administration of systemic penicillin [17]. However, though the incidence of syphilis in patients with scleritis has not been widely reported in the literature, cases are increasing in the United States, making it a priority to include this condition in initial workups of patients with scleritis [18].

A statistical analysis of independent variables such as age, sex, laterality of scleritis, type of scleritis, and ocular complications as risk factors for associated immune-mediated or infectious diseases was performed. Nodular anterior scleritis was statistically significantly less likely to present in patients with immune-mediated disease. Akpek et al.’s series found that men were more likely than women to have an infectious disease, while women were more likely than men to have an immune-mediated disease; we did not see this pattern in our series [4]. Furthermore, nodular anterior scleritis was clearly associated with infection in Akpek et al.’s series, while in our series, it was less likely to be seen in the immune-mediated-disease group [4]. Our finding could be related to what was also reported by Jabs et al. series, in which nodular anterior scleritis seemed to be better controlled with NSAIDs than with immunosuppressive treatment. [2]. This treatment’s effectiveness could be because patients with nodular anterior scleritis are less likely to have an immune-mediated condition as the cause and would require immunosuppressive therapy to control the inflammation of the sclera.

Lastly, multiple cases of scleritis in patients prescribed medications, such as bisphosphonates, topiramate, and etanercept have been reported [19,20,21]. The mechanism for ocular inflammation in patients taking these medications is unknown but might be related to the activation of T cells that release cytokines that, in turn, contribute to an inflammatory eye response [19]. Our series includes one patient diagnosed with scleritis while receiving treatment with ATRA for acute promyelocytic leukemia (APL). While this association has not previously been reported, Newman et al. described two patients with APL who were treated with ATRA and diagnosed with differentiation syndrome, a fatal drug reaction [22]. Both patients had experienced a bilateral reduction of VA with multifocal serous retinal detachment, contributing to the idea that ATRA may be associated with ocular complications [22]. This discovery is important as it alerts physicians to check for possible medication-mediated scleritis in patients who, during treatment, experience a persistent decrease in vision, ocular pain, or both. If treated in time, complications, such as uveitis, glaucoma, retinal detachment, and perforation of the globe, can be avoided [21].

As with all retrospective studies, the data must be interpreted with caution as the nature of the study may have introduced ascertainment bias. The study was limited to two uveitis specialists’ practices, and referral bias may have been introduced. Because our study took place in clinics with uveitis specialists, its results may not accurately represent the general Hispanic population in Puerto Rico. It is also possible that more straightforward cases, particularly those associated with less severe systemic disease, may not have been referred. It is also possible that more straightforward cases, particularly those associated with less severe systemic disease, may have yet to be referred. Additionally, the number of patients in some groups, such as those with an associated infectious disease, is small, limiting statistical power for odds ratio analyses. Further studies should be performed to analyze the associations between systemic diseases and specific types of scleritis and response to treatment.

## Figures and Tables

**Table 1 jcm-12-01969-t001:** Characteristics of the patients with scleritis at presentation.

Characteristic	Results
Number of patients	141
Age	
Median (yrs.)	53
Range (yrs.)	10–86
Sex (percentage that are females)	70.21
Hispanic (percentage)	100
Ocular involvement (percentage unilateral)	73.80
Type of scleritis	
Diffuse anterior scleritis	72.34
Nodular anterior scleritis	19.14
Posterior scleritis	7.80
Necrotizing scleritis	4.30
Immune-mediated systemic disease (percentage of patients)	33.33
Rheumatoid arthritis	22.70
Sjögren’s syndrome	3.55
Relapsing polychondritis	2.87
Seronegative spondyloarthropathies	1.42
Sarcoidosis	1.42
Systemic lupus erythematosus	1.42
Systemic vasculitis	0.71
Other immune-mediated diseases	2.87
Infectious disease (percentage of patients)	5.70
Syphilis	2.13
Herpes simplex	1.41
Herpes zoster ophthalmicus	1.41
Lyme disease	0.71
Medication-associated scleritis	0.71

**Table 2 jcm-12-01969-t002:** Characteristics of eyes with scleritis.

Eye-Specific Characteristics	Results
Eyes with scleritis	178
Type of scleritis (percentage of eyes)	–
Diffuse anterior scleritis	71.9
Nodular anterior scleritis	18.0
Posterior scleritis	8.4
Necrotizing scleritis	5.1
Ocular complication (percentage of affected eyes)	–
Uveitis	13.5
Keratitis	1.7
Visual acuity equal to or greater than 20/50 (percentage of affected eyes)	77.0
Visual acuity equal to or less than 20/200 * (percentage of affected eyes)	11.8
IOP greater than 21 (percentage of affected eyes)	9.6

* Less than 20/200 includes counting fingers, hand motion, and no light perception; IOP: intraocular pressure.

**Table 3 jcm-12-01969-t003:** Risk Factors for infectious or immune-mediated disease.

Risk Factor	OR	95% CI	*p*
Infectious disease			
Age (≥53 vs. <53 yrs.)	0.45	0.10–1.96	0.299
Sex (female vs. male)	0.69	0.15–3.04	0.695
Laterality (bilateral vs. unilateral disease)	No bilateral cases		0.996
Scleritis type			
Diffuse anterior (yes vs. no)	2.8	0.33–23.53	0.443
Nodular anterior (yes vs. no)	1.44	0.27–7.56	0.649
Necrotizing (yes vs. no)	No cases		0.998
Posterior (yes vs. no)	No cases		0.998
Keratitis (any vs. none)	No cases		0.999
Uveitis (yes vs. no)	4.09	0.90–18.70	0.086
Immune-mediated disease			
Age (≥53 vs. <53 yrs.)	1.34	0.66–2.75	0.472
Sex (female vs. male)	1.54	0.69–3.44	0.331
Laterality (bilateral vs. unilateral disease)	1.88	0.86–4.09	0.152
Scleritis type			
Diffuse anterior (yes vs. no)	2.30	0.96–5.52	0.071
Nodular anterior (yes vs. no)	0.21	0.06–0.73	0.011
Necrotizing (yes vs. no)	2.13	0.41–11.04	0.391
Posterior (yes vs. no)	1.81	0.52–6.27	0.339
Keratitis (any vs. none)	1.03	0.09–11.70	1.00
Uveitis (yes vs. no)	1.46	0.55–3.85	0.451

CI: confidence interval; OR: odds ratio.

## Data Availability

The data presented in this study are available on request from the corresponding author.

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
