# Peer review of "Systemic Disease Associations in a Cohort of Hispanic Patients with Scleritis"

_jcm, 2023, doi:10.3390/jcm12051969_

Round 1
Reviewer 1 Report
1. In the abstract, the part of Conslusion, the authours draw the conclusion that patients with nodular scleritis have a lower risk of having an associated immune mediated disease, thus the data that supporting this conclusion should also be included in the part of Result.
2.What is the diagnosis standard for immune disease or infectious disease should also be clarified in the part of Methods, for example of Herpes simplex.
Author Response
Response to Reviewer 1 Comments
Point 1: In the abstract, the part of Conslusion, the authours draw the conclusion that patients with nodular scleritis have a lower risk of having an associated immune mediated disease, thus the data that supporting this conclusion should also be included in the part of Result.
Response 1:
Thank you for your thoughtful review of our manuscript. We appreciate your concern. The data supporting our conclusion is shown on table 3. Also, please note that it is stated in line 34-36 of the abstract (results) section and line 160-162 of the results section: The risk factors for infectious and immune-mediated diseases in patients with scleritis are shown in Table 3.
Line 34-36: “Statistical analysis revealed that patients with nodular anterior scleritis were less likely to have an associated immune-mediated disease (OR: 0.21; P = .011).”
Line 160-162: “Nodular anterior scleritis (OR = 0.21; P = .011) was the only type of scleritis seen less often in patients with an immune-mediated disease. No other significant risk factors were found.”
Point 2: What is the diagnosis standard for immune disease or infectious disease should also be clarified in the part of Methods, for example of Herpes simplex.
Response 2:
We appreciate your question and insight. As mentioned in the first paragraph of our methods section: “All the patients underwent a comprehensive physical examination and an extensive systemic workup to identify scleritis-associated conditions such as rheumatoid arthritis, systemic vasculitis, systemic lupus erythematosus (SLE), syphilis, and viral etiologies. The ascertainment of an associated immune-mediated or infectious disease at a given patient’s presentation was made by a review of that individual’s clinical history, an examination, and a clinical diagnosis supported by ancillary laboratory testing. Consultations by an internal medicine specialist or rheumatologist were sought when needed, on a case-by-case basis. The previously identified conditions were included in each patient’s history if that person had been diagnosed before the initial visit (within one month) or because of the initial medical workup order for the scleritis diagnosis.”
A more detailed discussion of all the potentially scleritis-associated immune-mediated and infectious diseases would be extensive and may fall beyond the scope of this manuscript.
Reviewer 2 Report
I want to thank the authors since they provided a precious insight to scleritis.
Correctly, they repeatedly said that their observation is limited to Hispanic patients only, so some caution should be adopted in inferring their conclusion to non-Hispanic patients.
Their retrospective study covers a 31-year long time span and this is significant
My suggestions to improve the manuscript.
-
provide some images of eyes with different types of scleritis
-
provide a Table with authors' suggestion for an everyday laboratory flow-chart in searching for associated diseases
Minor issues
Line 72: .. file records were related to scleritis patients from …
Line 102-3: why were patients with scleritis of unknown origin (primary scleritis) excluded form the study? How many patients (and eyes) were excluded?
Table 2 – 11.8% of eyes had BCVA ≤ 20/200 and at line 199-200 it is said that this was unrelated to scleritis. Maybe a few lines providing the causes of vision impairment would help
Author Response
Response to Reviewer 2 Comments
Point 1: Provide some images of eyes with different types of scleritis.
Response 1:
Thank you for your thoughtful review of our manuscript. We agree that it would be optimal to have pictures representative of different types of scleritis. Although we have many pictures, we feel that none are specific and illustrative as to serve a didactical purpose. We understand that our manuscript may be well understood without such illustrations; and that outstanding illustrations on the different types of scleritis are readily available throughout the internet and within many excellent review articles on the subject.
Point 2: Provide a Table with authors' suggestion for an everyday laboratory flow-chart in searching for associated diseases.
Response 2:
We appreciate your kind suggestion regarding flow-chart for everyday laboratory to search associated diseases. However, we believe that this would be beyond the scope of our present manuscript. We will strongly consider a review article focusing on an algorithmic approach to diagnosis scleritis-associated conditions.
Minor issues:
Point 1: Line 72: .. file records were related to scleritis patients from …
Response 1:
As suggested, we changed the sentence to the following: “We performed a retrospective review of medical records generated from January 1990 through July 2021; file records were related to scleritis patients from two private uveitis practices and the pertinent outpatient clinics of the Department of Ophthalmology of the University of Puerto Rico School of Medicine.”
Point 2: Line 102-3: why were patients with scleritis of unknown origin (primary scleritis) excluded from the study? How many patients (and eyes) were excluded?
Response 2:
We appreciate your question. In fact, patients with primary scleritis were not excluded. As shown in Table 1 from a total of 141 patients, 33.33% had an immune-mediated systemic disease and 5.70 had an associated infectious etiology. The remainder 60.97% had primary scleritis. This also presented in the third paragraph of the results section. The last sentence in line 101-102 stating that the patients with an unknown type of scleritis were excluded from the study, was deleted accordingly.
Point 3: Table 2 – 11.8% of eyes had BCVA ≤ 20/200 and at line 199-200 it is said that this was unrelated to scleritis. Maybe a few lines providing the causes of vision impairment would help.
Response 3:
Thank you for your suggestion. However, line 199-200 what the author meant was that our study suggests that in patients with scleritis, poor vision upon presentation, only occurred in a minimal portion of patients, regardless of cause.